# GRAM: Graph-based Attention Model for Healthcare Representation Learning

**Edward Choi[1], Mohammad Taha Bahadori[1], Le Song[1], Walter F. Stewart[2] & Jimeng Sun[1]**
[1]Georgia Institute of Technology,      [2]Sutter Health

## Abstract

Deep learning methods exhibit promising performance for predictive modeling in healthcare, but two important challenges remain:

- *Data insufficiency:* Often in healthcare predictive modeling, the sample size is insufficient for deep learning methods to achieve satisfactory results.
- *Interpretation:* The representations learned by deep learning models should align with medical knowledge.

To address these challenges, we propose a GRaph-based Attention Model, GRAM that supplements electronic health records (EHR) with hierarchical information inherent to medical ontologies. Based on the data volume and the ontology structure, GRAM represents a medical concept as a combination of its ancestors in the ontology via an attention mechanism.

We compared predictive performance (*i.e.* accuracy, data needs, interpretability) of GRAM to various methods including the recurrent neural network (RNN) in two sequential diagnoses prediction tasks and one heart failure prediction task. Compared to the basic RNN, GRAM achieved 10% higher accuracy for predicting diseases rarely observed in the training data and 3% improved area under the ROC curve for predicting heart failure using an order of magnitude less training data. Additionally, unlike other methods, the medical concept representations learned by GRAM are well aligned with the medical ontology. Finally, GRAM exhibits intuitive attention behaviors by adaptively generalizing to higher level concepts when facing data insufficiency at the lower level concepts.

## 1 Introduction

The rapid growth in volume and diversity of health care data from electronic health records (EHR) and other sources is motivating the use of predictive modeling to improve care for individual patients. In particular, novel applications are emerging that use deep learning methods such as word embedding (Choi et al., 2016c;e), recurrent neural networks (RNN) (Che et al., 2016; Choi et al., 2016a;b; Lipton et al., 2016), convolutional neural networks (CNN) (Nguyen et al., 2016) or stacked denoising autoencoders (SDA) (Che et al., 2015; Miotto et al., 2016), demonstrating significant performance enhancement for diverse prediction tasks. Deep learning models appear to perform significantly better than logistic regression or multilayer perceptron (MLP) models that depend, to some degree, on expert feature construction (Lipton et al., 2015; Razavian et al., 2016).

Training deep learning models typically requires large amounts of data that often cannot be met by a single health system or provider organization. Sub-optimal model performance can be particularly challenging when the focus of interest is predicting onset of a specific disease (e.g. heart failure) or related events such as accelerated disease progression. For example, using Doctor AI (Choi et al., 2016a), we discovered that RNN alone was ineffective to predict the onset of diseases such as cerebral degenerations (e.g. Leukodystrophy, Cerebral lipidoses) or developmental disorders (e.g. autistic disorder, Heller's syndrome), partly because their rare occurrence in the training data provided little learning opportunity to the flexible models like RNN.

The data requirement of deep learning models comes from having to assess exponential number of combinations of input features. This can be alleviated by exploiting medical ontologies that encodes hierarchical clinical constructs and relationships among medical concepts. Fortunately, there are many well-organized ontologies in healthcare such as the International Classification of Diseases (ICD), Clinical Classifications Software (CCS) (Stearns et al., 2001) or Systematized Nomenclature of Medicine-Clinical Terms (SNOMED-CT) (Project et al., 2010). Nodes (*i.e.* medical concepts) close to one another in medical ontologies are likely to be associated with similar patients, allowing

us to transfer knowledge among them. Therefore, proper use of medical ontologies will be helpful when we lack enough data for the nodes in the ontology to train deep learning models.

In this work, we propose GRAM, a method that infuses information from medical ontologies into deep learning models via neural attention. Considering the frequency of a medical concept in the EHR data and its ancestors in the ontology, GRAM decides the representation of the medical concept by adaptively combining its ancestors via attention mechanism. This will not only support deep learning models to learn robust representations without large amount of data, but also learn interpretable representations that align well with the knowledge from the ontology. The attention mechanism is trained in an end-to-end fashion with the neural network model that predicts the onset of disease(s). We also propose an effective initialization technique in addition to the ontological knowledge to better guide the representation learning process.

We compared predictive performance (i.e. accuracy, data needs, interpretability) of GRAM to various models including the recurrent neural network (RNN) in two sequential diagnoses prediction tasks and one heart failure (HF) prediction task. We demonstrate that GRAM is up to 10% more accurate than the basic RNN for predicting diseases less observed in the training data. After discussing GRAM's scalability, we visualize the representations learned from various models where GRAM provides more intuitive representations by grouping similar medical concepts close to one another. Finally, we show GRAM's attention mechanism can be interpreted to understand how it assigns the right amount of attention to the ancestors of each medical concept by considering the data availability and the ontology structure.

## 2 METHODOLOGY

We first define the notations describing EHR data and medical ontologies, followed by a description of GRAM (Section 2.2), the end-to-end training of the attention generation and predictive modeling (Section 2.3), and the efficient initialization scheme (Section 2.4).

### 2.1 BASIC NOTATION

We denote the set of entire medical codes from the EHR as $c_1, c_2, \ldots, c_{|\mathcal{C}|} \in \mathcal{C}$ with the vocabulary size $|\mathcal{C}|$. The clinical record of each patient can be viewed as a sequence of visits $V_1, \ldots, V_T$ where each visit contains a subset of medical codes $V_t \subseteq \mathcal{C}$. $V_t$ can be represented as a binary vector $\mathbf{x}_t \in \{0, 1\}^{|\mathcal{C}|}$ where the $i$-th element is 1 only if $V_t$ contains the code $c_i$. To avoid clutter, all algorithms will be presented for a single patient.

We assume that a given medical ontology $\mathcal{G}$ typically expresses the hierarchy of various medical concepts in the form of a *parent-child* relationship, where the medical codes $\mathcal{C}$ form the leaf nodes. Ontology $\mathcal{G}$ is represented as a directed acyclic graph (DAG) whose nodes form a set $\mathcal{D} = \mathcal{C} + \mathcal{C}'$. $\mathcal{C}' = \{c_{|\mathcal{C}|+1}, c_{|\mathcal{C}|+2}, \ldots, c_{|\mathcal{C}|+|\mathcal{C}'|}\}$ defines the set of all non-leaf nodes (*i.e.* ancestors of the leaf nodes), where $|\mathcal{C}'|$ represents the number of all non-leaf nodes. We use *knowledge DAG* to refer to $\mathcal{G}$. A parent in the knowledge DAG $\mathcal{G}$ represents a related but more general concept over its children. Therefore, $\mathcal{G}$ provides a multi-resolution view of medical concepts with different degrees of specificity. While some ontologies are exclusively expressed as parent-child hierarchies (e.g. ICD-9, CCS), others are not. For example, in some instances SNOMED-CT also links medical concepts to causal or treatment relationships, but the majority relationships in SNOMED-CT are still parent-child. Therefore, we focus on the parent-child relationships in this work.

### 2.2 KNOWLEDGE DAG AND THE ATTENTION MECHANISM

GRAM leverages the *parent-child* relationship of $\mathcal{G}$ to learn robust representations when data volume is constrained. GRAM balances the use of ontology information in relation to data volume in determining the level of specificity for a medical concept. When a medical concept is less observed in the data, more weight is given to its ancestors as they can be learned more accurately and offer general (coarse-grained) information about their children. The process of resorting to the parent concepts can be automated via the attention mechanism and the end-to-end training as described in Figure 1.

In the knowledge DAG, each node $c_i$ is assigned a basic embedding vector $\mathbf{e}_i \in \mathbb{R}^m$, where $m$ represents the dimensionality. Then $\mathbf{e}_1, \ldots, \mathbf{e}_{|\mathcal{C}|}$ are the basic embeddings of the codes $c_1, \ldots, c_{|\mathcal{C}|}$ while $\mathbf{e}_{|\mathcal{C}|+1}, \ldots, \mathbf{e}_{|\mathcal{C}|+|\mathcal{C}'|}$ represent the basic embeddings of the internal nodes $c_{|\mathcal{C}|+1}, \ldots, c_{|\mathcal{C}|+|\mathcal{C}'|}$. The initialization of these basic embeddings is described in Section 2.4. We formulate a leaf node's

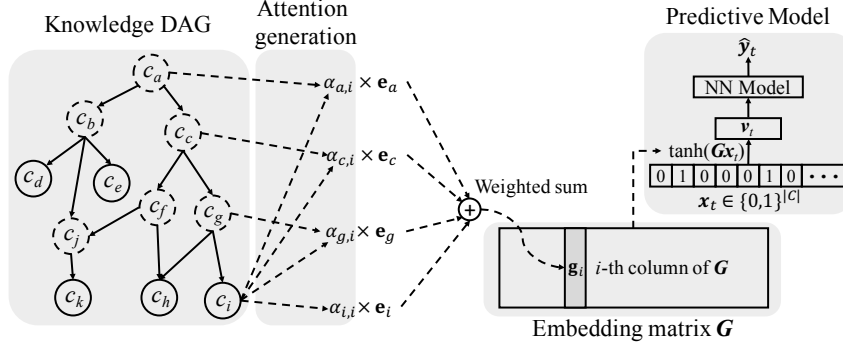

Figure 1: The illustration of GRAM. Leaf nodes (solid circles) represents a medical concept in the EHR, while the non-leaf nodes (dotted circles) represent more general concepts. The final representation $\mathbf{g}_i$ of the leaf concept $c_i$ is computed by combining the basic embeddings $\mathbf{e}_i$ of $c_i$ and $\mathbf{e}_g, \mathbf{e}_c$ and $\mathbf{e}_a$ of its ancestors $c_g, c_c$ and $c_a$ via an attention mechanism. The final representations form the embedding matrix $\mathbf{G}$ for all leaf concepts. After that, we use $\mathbf{G}$ to embed patient visit vector $\mathbf{x}_t$ to a visit representation $\mathbf{v}_t$, which is then fed to a neural network model to make the final prediction $\hat{\mathbf{y}}_t$.

final representation as a convex combination of the basic embeddings of itself and its ancestors:

$$\mathbf{g}_i = \sum_{j \in \mathcal{A}(i)} \alpha_{ij} \mathbf{e}_j, \qquad \sum_{j \in \mathcal{A}(i)} \alpha_{ij} = 1, \ \ \alpha_{ij} \geq 0 \ \text{ for } j \in \mathcal{A}(i), \tag{1}$$

where $\mathbf{g}_i \in \mathbb{R}^m$ denotes the final representation of the code $c_i$, $\mathcal{A}(i)$ the indices of the code $c_i$ and $c_i$'s ancestors, $\mathbf{e}_j$ the basic embedding of the code $c_j$ and $\alpha_{ij} \in \mathbb{R}$ the attention weight on the embedding $\mathbf{e}_j$ when calculating $\mathbf{g}_i$. The attention weight $\alpha_{ij}$ in Eq. (1) is calculated by the following Softmax function,

$$\alpha_{ij} = \frac{\exp(f(\mathbf{e}_i, \mathbf{e}_j))}{\sum_{k \in \mathcal{A}(i)} \exp(f(\mathbf{e}_i, \mathbf{e}_k))} \tag{2}$$

$f(\mathbf{e}_i, \mathbf{e}_j)$ is a scalar value representing the compatibility between the basic embeddings of $\mathbf{e}_i$ and $\mathbf{e}_k$. We compute $f(\mathbf{e}_i, \mathbf{e}_j)$ via the following feed-forward network with a single hidden layer (MLP),

$$f(\mathbf{e}_i, \mathbf{e}_j) = \mathbf{u}_a^\top \tanh\left(\mathbf{W}_a \begin{bmatrix} \mathbf{e}_i \\ \mathbf{e}_j \end{bmatrix} + \mathbf{b}_a\right) \tag{3}$$

where $\mathbf{W}_a \in \mathbb{R}^{l \times 2m}$ is the weight matrix for the concatenation of $\mathbf{e}_i$ and $\mathbf{e}_j$, $\mathbf{b} \in \mathbb{R}^l$ the bias vector, and $\mathbf{u}_a \in \mathbb{R}^l$ the weight vector for generating the scalar value. The constant $l$ represents the dimension size of the hidden layer of $f(\cdot, \cdot)$. Note that we always concatenate $\mathbf{e}_i$ and $\mathbf{e}_j$ in the child-ancestor order.

**Remarks:** The example in Figure 1 is derived based on a single path from $c_i$ to $c_a$. However, the same mechanism can be applicable to multiple paths as well. For example, code $c_k$ has two paths to the root $c_a$, containing five ancestors in total. Another scenario is where the EHR data contain both leaf codes and some ancestor codes. We can move those ancestors present in EHR data from the set $\mathcal{C}'$ to $\mathcal{C}$ and apply the same process as Eq. (1) to obtain the final representations for them.

## 2.3 End-to-End Training with a Predictive Model

We train the attention mechanism together with a predictive model such that the attention mechanism improves the predictive performance. Once the final representations $\mathbf{g}_1, \mathbf{g}_2, \ldots, \mathbf{g}_{|\mathcal{C}|}$ of all medical codes are obtained, we can convert visit $V_t$ to a visit representation $\mathbf{v}_t$ by using the embedding matrix $\mathbf{G} \in \mathcal{R}^{m \times |\mathcal{C}|}$ where $\mathbf{g}_i$ is its $i$-th column as in Figure 1. We continue the mathematical formulation under the assumption that we are using the RNN to perform sequential diagnoses prediction (Choi et al., 2016a;b) with the objective of predicting the disease codes of the next visit $V_{t+1}$ given the visit records up to the current timestep $V_1, V_2, \ldots, V_t$, which can be expressed as follows,

$$\hat{\mathbf{y}}_t = \hat{\mathbf{x}}_{t+1} = \text{Softmax}(\mathbf{W}\mathbf{h}_t + \mathbf{b}), \quad \text{where}$$
$$\mathbf{h}_1, \mathbf{h}_2, \ldots, \mathbf{h}_t = \text{RNN}(\mathbf{v}_1, \mathbf{v}_2, \ldots, \mathbf{v}_t), \quad \text{where} \tag{4}$$
$$\mathbf{v}_1, \mathbf{v}_2, \ldots, \mathbf{v}_t = \tanh(\mathbf{G}[\mathbf{x}_1, \mathbf{x}_2, \ldots, \mathbf{x}_t])$$

Table 1: Basic statistics of Sutter PAMF, MIMIC-III and Sutter heart failure (HF) cohort.

| Dataset | Sutter PAMF | MIMIC-III | Sutter HF cohort |
|---|---|---|---|
| # of patients | 258,555[†] | 7,499[†] | 30,727[†] (3,408 cases) |
| # of visits | 13,920,759 | 19,911 | 572,551 |
| Avg. # of visits per patient | 53.8 | 2.66 | 38.38 |
| # of unique ICD9 codes | 10,437 | 4,893 | 5,689 |
| Avg. # of codes per visit | 1.98 | 13.1 | 2.06 |
| Max # of codes per visit | 54 | 39 | 29 |

† Note that for all datasets, we selected patients who made at least two hospital visits.

where $\mathbf{x}_t \in \mathbb{R}^{|\mathcal{C}|}$ denotes the $t$-th visit; $\mathbf{v}_t \in \mathbb{R}^m$ the $t$-th visit representation; $\mathbf{h}_t \in \mathbb{R}^r$ the RNN's hidden layer at $t$-th time step (*i.e.* $t$-th visit); $\mathbf{W} \in \mathbb{R}^{|\mathcal{C}| \times r}$ and $\mathbf{b} \in \mathbb{R}^{|\mathcal{C}|}$ the weight matrices and the bias vector of the Softmax function; $r$ denotes the dimension size of the hidden layer. We use "RNN" to denote any recurrent neural network variants that can cope with the vanishing gradient problem (Bengio et al., 1994), such as LSTM (Hochreiter & Schmidhuber, 1997), GRU (Cho et al., 2014), and IRNN (Le et al., 2015), with any varying numbers of hidden layers. The prediction loss for all time steps is calculated using the cross entropy as follows, $\mathcal{L}(\mathbf{x}_1, \mathbf{x}_2 \dots, \mathbf{x}_T) = -\frac{1}{T-1} \sum_{t=1}^{T-1} \left( \mathbf{y}_t^\top \log(\widehat{\mathbf{y}}_t) + (1 - \mathbf{y}_t)^\top \log(1 - \widehat{\mathbf{y}}_t) \right)$ where we sum the cross entropy errors from all dimensions of $\widehat{\mathbf{y}}_t$, $T$ denotes the length of the visit sequence. Note that the above loss is defined for a single patient. But we can take the average of the individual loss for multiple patients.

## 2.4 INITIALIZING BASIC EMBEDDINGS

The attention generation mechanism in Section 2.2 requires basic embeddings $\mathbf{e}_i$ of each node in the knowledge DAG. The basic embeddings of ancestors, however, pose a difficulty because they are often not observed in the data.To better initialize them, we use co-occurrence information to learn the basic embeddings of medical codes and their ancestors. Co-occurrence has proven to be an important source of information when learning representations of words or medical concepts (Mikolov et al., 2013; Choi et al., 2016c;e). To train the basic embeddings, we employ GloVe (Pennington et al., 2014), which uses the global co-occurrence matrix of words to learn their representations. In our case, the co-occurrence matrix of the codes and the ancestors was generated by counting the co-occurrences within each visit $V_t$, where we augment each visit with the ancestors of the codes in the visit. Details of training the basic embeddings are described in the Appendix A. Note that, with or without the initialization, the basic embeddings $\mathbf{e}_i$'s of both leaf nodes (*i.e.* medical codes) and non-leaf nodes (*i.e.* ancestors) are fine-tuned when training our model, since the error signal flows from the output $\widehat{\mathbf{y}}_t$ to the final representations $\mathbf{g}_i$'s which are convex combinations of $\mathbf{e}_i$'s.

## 3 EXPERIMENTS

We conduct three experiments to determine if GRAM offered superior prediction performance when facing data insufficiency. We first describe the experimental setup followed by results comparing predictive performance of GRAM with various baseline models. After discussing GRAM's scalability, we qualitatively evaluate the interpretability of the resulting representation. The source code of GRAM is publicly available at `https://github.com/mp2893/gram`.

## 3.1 EXPERIMENT SETUP

**Prediction tasks and source of data:** We conduct two sequential diagnoses prediction tasks, which aim at predicting all diagnosis categories in the next visit, and one heart failure (HF) prediction task, which is a binary prediction task for predicting a future HF onset where the prediction is made only once at the last visit $\mathbf{x}_T$.

Two sequential diagnoses predictions are respectively conducted using 1) Sutter Palo Alto Medical Foundation (PAMF) dataset, which consists of 18-years longitudinal medical records of 258K patients between age 50 and 90. This will determine GRAM's performance for general adult population with long visit records. 2) MIMIC-III dataset (Johnson et al., 2016; Goldberger et al., 2000), which is a publicly available dataset consisting of medical records of 7.5K intensive care unit (ICU) patients over 11 years. This will determine GRAM's performance for high-risk patients with very short visit records. We utilize all the patients with at least 2 visits. We prepared the true labels $\mathbf{y}_t$ by grouping the ICD9 codes into 283 groups using CCS single-level diagnosis grouper[1]. This is to improve the

---

[1]https://www.hcup-us.ahrq.gov/toolssoftware/ccs/AppendixASingleDX.txt

training speed and predictive performance for easier analysis, while preserving sufficient granularity for each diagnosis. Each diagnosis code's varying frequency in the training data can be viewed as different degrees of data insufficiency. We calculate *Accuracy@k* for each of CCS single-level diagnosis codes such that, given a visit $V_t$, we get 1 if the target diagnosis is in the top $k$ guesses and 0 otherwise.

We conduct HF prediction on Sutter heart failure (HF) cohort, which is a subset of Sutter PAMF data for a heart failure onset prediction study with 3.4K HF cases and 27K controls chosen by a set of criteria (see Appendix B). This will determine GRAM's performance for a different prediction task where we predict the onset of one specific condition. We randomly downsample the training data to create different degrees of data insufficiency. We use area under the ROC curve (AUC) to measure the performance.

A summary of the datasets are provided in Table 1.We used CCS multi-level diagnoses hierarchy[2] as our knowledge DAG $\mathcal{G}$. We also tested the ICD9 code hierarchy[3], but the performance was similar to using CCS multi-level hierarchy. For all three tasks, we randomly divide the dataset into the training, validation and test set by .75:.10:.15 ratio, and use the validation set to tune the hyper-parameters. Further details regarding the hyper-parameter tuning are provided in Appendix C. The test set performance is reported in the paper.

**Implementation details:** We implemented GRAM with Theano 0.8.2 (Team, 2016). For training models, we used Adadelta (Zeiler, 2012) with a mini-batch of 100 patients, on a machine equipped with Intel Xeon E5-2640, 256GB RAM, four Nvidia Titan X's and CUDA 7.5.

**Models for comparison** are the following. The first two GRAM+ and GRAM are the proposed methods and the rest are baselines. Hyper-parameter tuning is configured so that the number of parameters for the baselines would be comparable to GRAM's. Further details are provided in Appendix C.

- **GRAM:** Input sequence $\mathbf{x}_1, \ldots, \mathbf{x}_T$ is first transformed by the embedding matrix $\mathbf{G}$, then fed to the GRU with a single hidden layer, which in turn makes the prediction, as described by Eq. (4). The basic embeddings $\mathbf{e}_i$'s are randomly initialized.

- **GRAM+:** We use the same setup as **GRAM**, but the basic embeddings $\mathbf{e}_i$'s are initialized according to Section 2.4.

- **RandomDAG:** We use the same setup as **GRAM**, but each leaf concept has five randomly assigned ancestors from the CCS multi-level hierarchy to test the effect of correct domain knowledge.

- **RNN**: Input $\mathbf{x}_t$ is transformed by an embedding matrix $\mathbf{W}_{emb} \in \mathbb{R}^{k \times |\mathcal{C}|}$, then fed to the GRU with a single hidden layer. The embedding size $k$ is a hyper-parameter. $\mathbf{W}_{emb}$ is randomly initialized and trained together with the GRU.

- **RNN+:** We use the same setup as **RNN**, but we initialize the embedding matrix $\mathbf{W}_{emb}$ with GloVe vectors trained only with the co-occurrence of leaf concepts. This is to compare GRAM with a similar weight initialization technique.

- **SimpleRollUp:** We use the same setup as **RNN**. But for input $\mathbf{x}_t$, we replace all diagnosis codes with their direct parent codes in the CCS multi-level hierarchy, giving us 578, 526 and 517 input codes respectively for Sutter data, MIMIC-III and Sutter HF cohort. This is to compare the performance of GRAM with a common grouping technique.

- **RollUpRare:** We use the same setup as **RNN**, but we replace any diagnosis code whose frequency is less than a certain threshold in the dataset with its direct parent. We set the threshold to 100 for Sutter data and Sutter HF cohort, and 10 for MIMIC-III, giving us 4,408, 935 and 1,538 input codes respectively for Sutter data, MIMIC-III and Sutter HF cohort. This is an intuitive way of dealing with infrequent medical codes.

## 3.2 PREDICTION PERFORMANCE AND SCALABILITY

Tables 2a and 2b show the sequential diagnoses prediction performance on Sutter data and MIMIC-III. Both figures show that GRAM+ outperforms other models when predicting labels with significant data insufficiency (*i.e.* less observed in the training data).The performance gain is greater for MIMIC-III, where GRAM+ outperforms the basic RNN by 10% in the 20th-40th percentile range. This seems to come from the fact that MIMIC patients on average have significantly shorter visit history than Sutter patients, with much more codes received per visit. Such short sequences make it difficult for the RNN to learn and predict diagnoses sequence. The performance difference between GRAM+ and

---

[2]https://www.hcup-us.ahrq.gov/toolssoftware/ccs/AppendixCMultiDX.txt
[3]http://www.icd9data.com/2015/Volume1/default.htm

Table 2: Performance of three prediction tasks. The x-axis of (a) and (b) represents the labels grouped by the percentile of their frequencies in the training data in non-decreasing order. For (c), we vary the size of the training data to train the models. (b) uses *Accuracy@20* because MIMIC-III has a large average number of codes per visit (see Table 1).

| Model | 0-20 | 20-40 | 40-60 | 60-80 | 80-100 |
|---|---|---|---|---|---|
| GRAM+ | **0.0150** | **0.3242** | 0.4325 | 0.4238 | 0.4903 |
| GRAM | 0.0042 | 0.2987 | 0.4224 | 0.4193 | 0.4895 |
| RandomDAG | 0.0050 | 0.2700 | 0.4010 | 0.4059 | 0.4853 |
| RNN+ | 0.0069 | 0.2742 | 0.4140 | 0.4212 | **0.4959** |
| RNN | 0.0080 | 0.2691 | 0.4134 | 0.4227 | 0.4951 |
| SimpleRollUp | 0.0085 | 0.3078 | **0.4369** | **0.4330** | 0.4924 |
| RollUpRare | 0.0062 | 0.2768 | 0.4176 | 0.4226 | 0.4956 |

(a) *Accuracy@5* of sequential diagnoses prediction on Sutter data

| Model | 0-20 | 20-40 | 40-60 | 60-80 | 80-100 |
|---|---|---|---|---|---|
| GRAM+ | **0.0672** | **0.1787** | **0.2644** | 0.2490 | 0.6267 |
| GRAM | 0.0556 | 0.1016 | 0.1935 | 0.2296 | 0.6363 |
| RandomDAG | 0.0329 | 0.0708 | 0.1346 | 0.1512 | 0.4494 |
| RNN+ | 0.0454 | 0.0843 | 0.2080 | 0.2494 | 0.6239 |
| RNN | 0.0454 | 0.0731 | 0.1804 | 0.2371 | 0.6243 |
| SimpleRollUp | 0.0578 | 0.1328 | 0.2455 | **0.2667** | **0.6387** |
| RollUpRare | 0.0454 | 0.0653 | 0.1843 | 0.2364 | 0.6277 |

(b) *Accuracy@20* of sequential diagnoses prediction on MIMIC-III

| Model | 10% | 20% | 30% | 40% | 50% | 60% | 70% | 80% | 90% | 100% |
|---|---|---|---|---|---|---|---|---|---|---|
| GRAM+ | 0.7970 | **0.8223** | 0.8307 | **0.8332** | **0.8389** | **0.8404** | **0.8452** | **0.8456** | **0.8447** | **0.8448** |
| GRAM | **0.7981** | 0.8217 | **0.8340** | **0.8332** | 0.8372 | 0.8377 | 0.8440 | 0.8431 | 0.8430 | 0.8447 |
| RandomDAG | 0.7644 | 0.7882 | 0.7986 | 0.8070 | 0.8143 | 0.8185 | 0.8274 | 0.8312 | 0.8254 | 0.8226 |
| RNN+ | 0.7930 | 0.8117 | 0.8162 | 0.8215 | 0.8261 | 0.8333 | 0.8343 | 0.8353 | 0.8345 | 0.8335 |
| RNN | 0.7811 | 0.7942 | 0.8066 | 0.8111 | 0.8156 | 0.8207 | 0.8258 | 0.8278 | 0.8297 | 0.8314 |
| SimpleRollUp | 0.7799 | 0.8022 | 0.8108 | 0.8133 | 0.8177 | 0.8207 | 0.8223 | 0.8272 | 0.8269 | 0.8258 |
| RollUpRare | 0.7830 | 0.8067 | 0.8064 | 0.8119 | 0.8211 | 0.8202 | 0.8262 | 0.8296 | 0.8307 | 0.8291 |

(c) AUC of HF onset prediction on Sutter HF cohort

Table 3: Scalablity result in per epoch training time in second (the number of epochs needed).

| **Model** | Sequential diagnosis prediction (Sutter data) | Sequential diagnosis prediction (MIMIC-III) | HF prediction (Sutter HF cohort) |
|---|---|---|---|
| GRAM | 525s (39 epochs) | 2s (11 epochs) | 12s (7 epochs) |
| RNN | 352s (24 epochs) | 1s (6 epochs) | 8s (5 epochs) |

GRAM suggests that our proposed initialization scheme of the basic embeddings $e_i$ is important for sequential diagnosis prediction.

Table 2c shows the HF prediction performance on Sutter HF cohort. GRAM and GRAM+ consistently outperforms other baselines (except RNN+) by 3∼4% AUC, and RNN+ by maximum 1.8% AUC. These differences are quite significant given that the AUC is already in the mid-80s, a high value for HF prediction, cf. (Choi et al., 2016d). Note that, for GRAM+ and RNN+, we used the downsampled training data to initialize the basic embeddings $e_i$'s and the embedding matrix $W_{emb}$ with GloVe, respectively. The result shows that the initialization scheme of the basic embeddings in GRAM+ gives limited improvement over GRAM. This stems from the different natures of the two prediction tasks. While the goal of HF prediction is to predict a binary label for the entire visit sequence, the goal of sequential diagnosis prediction is to predict the co-occurring diagnosis codes at every visit. Therefore the co-occurrence information infused by the initialized embedding scheme is more beneficial to sequential diagnosis prediction. Additionally, this benefit is associated with the natures of the two prediction tasks than the datasets used for the prediction tasks. Because the initialized embedding shows different degrees of improvement as shown by Tables 2a and 2c, when Sutter HF cohort is a subset of Sutter PAMF, thus having similar characteristics. Additional prediction results when varying the $k$ of *Accuracy@k* are discussed in the Appendix D.

Overall, GRAM showed superior predictive performance under data insufficiency in three different experiments, demonstrating its general applicability in predictive healthcare modeling. Now we briefly discuss the scalability of GRAM by comparing its training time to RNN's. Table 3 shows the number of seconds taken for the two models to train for a single epoch for each predictive modeling task. GRAM+ and RNN+ showed the same behavior as GRAM and RNN. GRAM takes approximately 50% more time to train for a single epoch for all prediction tasks. This stems from calculating attention weights and the final representations $g_i$ for all medical codes. GRAM also generally takes about 50% more epochs to reach to the model with the lowest validation loss. This is due to optimizing an extra MLP model that generates the attention weights. Overall, use of GRAM adds a manageable amount of overhead in training time to the plain RNN.

### 3.3 QUALITATIVE EVALUATION OF INTERPRETABLE REPRESENTATIONS

To qualitatively assess the interpretability of the learned representations of the medical codes, we plot on a 2-D space using t-SNE (Maaten & Hinton, 2008) the final representations $g_i$ of 2,000

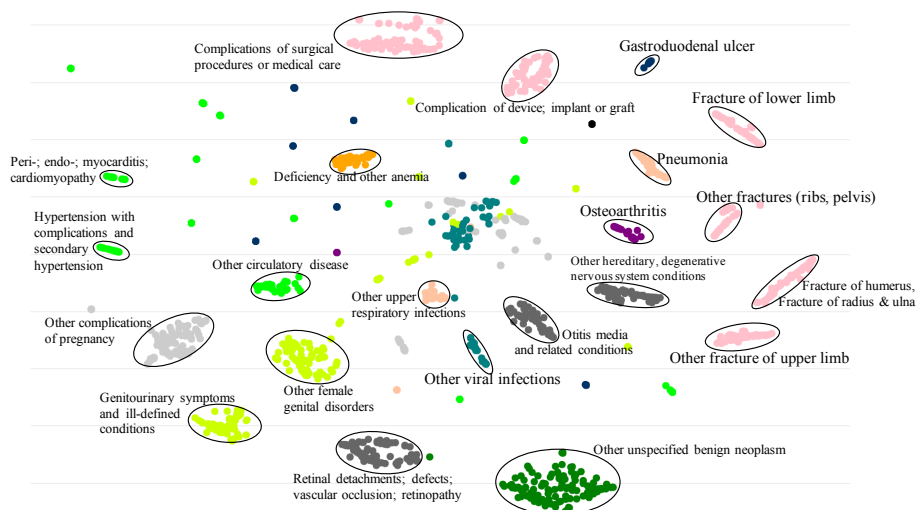

(a) Scatterplot of the final representations $\mathbf{g}_i$'s of GRAM+

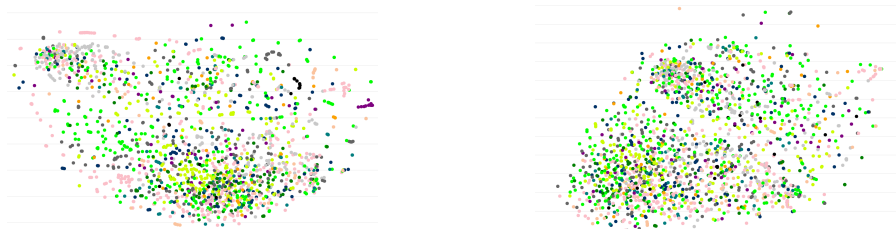

(b) Scatterplot of the trained embedding matrix $\mathbf{W}_{emb}$ of RNN+

(c) Scatterplot of the disease representations trained by GloVe

Figure 2: t-SNE scatterplots of medical concepts trained by GRAM+, RNN+ and GloVe

randomly chosen diseases learned by GRAM+ for sequential diagnoses prediction on Sutter data[4] (Figure 2a). The colors represent the highest disease categories and the text annotations represent the detailed disease categories in CCS multi-level hierarchy. For comparison, we also show the t-SNE plots on the strongest results from RNN+ (Figure 2b), and GloVe (Figure 2c), the same embedding technique in initializing the basic embeddings $\mathbf{e}_i$. Figures 2b and 2c confirm that interpretable representations cannot simply be learned only by co-occurrence or supervised prediction without medical knowledge. GRAM+ learns disease representations that are significantly more consistent with the given knowledge DAG $\mathcal{G}$. Therefore the neural network predictive model that accepts $\mathbf{g}_i$ is using accurate representations that lead to higher predictive performance. Additional scatterplots of other models are provided in Appendix E for comparison. An interactive visualization tool can be accessed at http://www.sunlab.org/research/gram-graph-based-attention-model/.

## 3.4 ANALYSIS OF THE ATTENTION BEHAVIOR

Next we show that GRAM's attention can be interpreted to understand how it considers data availability and knowledge DAG's structure when performing a prediction task. Using Eq. (1), we can calculate the attention weights of individual disease. Figure 3 shows the attention behaviors of four representative diseases when performing HF prediction on Sutter HF cohort.

*Other pneumothorax* (ICD9 512.89) in Figure 3a is rarely observed in the data and has only five siblings. In this case, most information is derived from the highest ancestor. *Temporomandibular joint disorders & articular disc disorder* (ICD9 524.63) in Figure 3b is rarely observed but has 139 siblings. In this case, its parent receives a stronger attention because it aggregates sufficient samples from all of its children to learn a more accurate representation. Note that the disease itself also receives a stronger attention to facilitate easier distinction from its large number of siblings.

---

[4]The scatterplots of models trained for sequential diagnoses prediction on MIMIC-III and HF prediction for Sutter HF cohort were similar but less structured due to smaller data size.

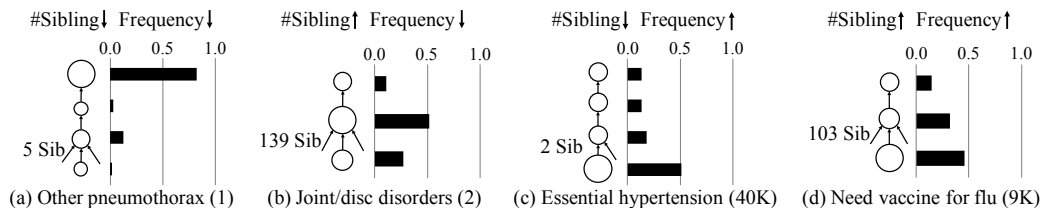

Figure 3: `GRAM`'s attention behavior during HF prediction for four representative diseases (each column). In each figure, the leaf node represents the disease and upper nodes are its ancestors. The size of the node shows the amount of attention it receives, which is also shown by the bar charts. The number in the parenthesis next to the disease is its frequency in the training data. We exclude the root of the knowledge DAG $\mathcal{G}$ from all figures as it did not play a significant role.

*Unspecified essential hypertension* (ICD9 401.9) in Figure 3c is very frequently observed but has only two siblings. In this case, `GRAM` assigns a very strong attention to the leaf, which is logical because the more you observe a disease, the stronger your confidence becomes. *Need for prophylactic vaccination and inoculation against influenza* (ICD9 V04.81) in Figure 3d is quite frequently observed and also has 103 siblings. The attention behavior in this case is quite similar to the case with fewer siblings (Figure 3b) with a slight attention shift towards the leaf concept as more observations lead to higher confidence.

## 4   RELATED WORK

We introduce recent studies related to `GRAM` that learn the representations of graphs and discuss their relationship with `GRAM`. Several studies focused on learning the representations of graph vertices by using the neighbor information. DeepWalk (Perozzi et al., 2014) and node2vec (Grover & Leskovec, 2016) use random walk while LINE (Tang et al., 2015) uses breadth-first search to find the neighbors of a vertex and learn its representation based on the neighbor information. Graph convolutional approaches (Yang et al., 2016; Kipf & Welling, 2016) also focus on learning the vertex representations to mainly perform vertex classification. These works focus on solving the graph data problems whereas `GRAM` focuses on solving EHR data problems using the knowledge DAG as supplementary information.

Several researchers tried to model the knowledge DAG such as WordNet (Miller, 1995) or Freebase (Bollacker et al., 2008) where two entities are connected with various types of relation, forming a set of triples. They aim to project entities and relations (Bordes et al., 2013; Socher et al., 2013; Wang et al., 2014; Lin et al., 2015) to the latent space based on the triples or additional information such as hierarchy of entities (Xie et al., 2016). These works demonstrated tasks such as link prediction, triple classification or entity classification using the learned representations. More recently, Li et al. (2016) learned the representations of words and Wikipedia categories by utilizing the hierarchy of Wikipedia categories. `GRAM` is fundamentally different from the above studies in that it aims to design intuitive attention mechanism on the knowledge DAG as a knowledge prior to cope with data insufficiency and learn medically interpretable representations to make accurate predictions.

A classical approach for incorporating side information in the predictive models is to use graph Laplacian regularization (Weinberger et al., 2006; Che et al., 2015). However, using this approach is not straightforward as it relies on the appropriate definition of distance on graphs which is often unavailable.

## 5   CONCLUSION

Data insufficiency, either due to less common diseases or small datasets, is one of the key hurdles in healthcare analytics, especially when we apply deep neural networks models. To overcome this hurdle, we leverage the knowledge DAG, which provides a multi-resolution view of medical concepts. We propose `GRAM`, a graph-based attention model using both a knowledge DAG and EHR to learn an accurate and interpretable representations for medical concepts. `GRAM` chooses a weighted average of ancestors of a medical concept and train the entire process with a predictive model in an end-to-end fashion. We conducted three predictive modeling experiments on real EHR datasets and showed significant improvement in the prediction performance, especially on low-frequency diseases and small datasets. Analysis of the attention behavior provided intuitive insight of `GRAM`.

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

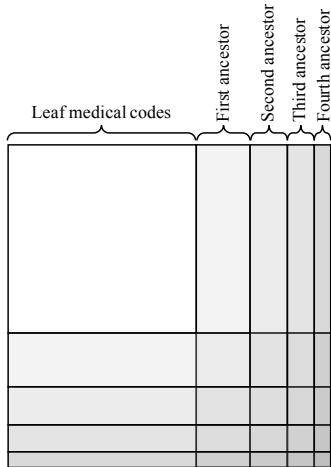

Figure 4: Creating the co-occurrence matrix together with the ancestors. Here we exclude the root node, which will be just a single row (column). We first create an augmented dataset by adding the ancestors of the code to the dataset. Then, we count the co-occurrence of the codes. Performing GloVe on this matrix produces the embedding vectors $\mathbf{e}_i$.

## A  GENERATING GLOVE EMBEDDINGS

We learn the basic embeddings $\mathbf{e}_i$'s of medical codes and their ancestors using GloVe (Pennington et al., 2014), which uses global co-occurrence matrix of words to learn their representations. We generate the co-occurrence matrix of the codes and the ancestors by counting the co-occurrence within each visit $V_t$. However, since visits only contain the *leaf* codes $c \in \mathcal{C}$, we augment each visit with the ancestors of the codes in each visit, then count the co-occurrence of codes and ancestors altogether.

We describe the details of the algorithm with an example. We borrow the parent-child relationships from the knowledge DAG of Figure 1. Given a visit $V_t$,

$$V_t = \{c_d, c_i, c_k\} \tag{5}$$

we augment it with the ancestors of all the codes to obtain the augmented visit $V_t'$,

$$V_t' = \{c_d, \underline{c_b}, \underline{c_a}, c_i, \underline{c_g}, \underline{c_c}, \underline{c_a}, c_k, \underline{c_j}, \underline{c_f}, \underline{c_c}, \underline{c_b}, \underline{c_a}\} \tag{6}$$

where the added ancestors are underlined. Note that a single ancestor can appear multiple times in $V_t'$. In fact, the higher the ancestor is in the knowledge DAG, the more times it is likely to appear in $V_t'$. We count the co-occurrence of two codes in $V_t'$ as follows,

$$co\text{-}occurrence(c_i, c_j, V_t') = count(c_i, V_t') \times count(c_j, V_t') \tag{7}$$

where $count(c_i, V_t')$ is the number of times the code $c_i$ appears in the augmented visit $V_t'$. For example, the co-occurrence between the leaf code $c_i$ and the root $c_a$ is 3. However, the co-occurrence between the ancestor $c_c$ and the root $c_a$ is 6. Therefore our algorithm will naturally make the ancestor codes have higher co-occurrence with other codes compared to leaf medical codes. We repeat this calculation for all pairs of codes in all augmented visits of all patients to obtain the co-occurrence matrix depicted by Figure 4. For training the embedding vectors using the co-occurrence matrix, we use the same procedure and hyper-parameter as described in Pennington et al. (2014).

## B  HEART FAILURE COHORT CONSTRUCTION

For the heart failure (HF) case patients, we select patients between 40 to 85 years of age at the time of HF diagnosis. HF diagnosis (HFDx) criteria are defined as: 1) Qualifying ICD-9 codes for HF appeared in the encounter records or medication orders. Qualifying ICD-9 codes are listed in Table 4. 2) at least three clinical encounters with qualifying ICD-9 codes had to occur within 12 months of each other, where the date of HFDx was assigned to the earliest of the three dates. If the time span between the first and second appearances of the HF diagnosis code was greater than 12 months, the date of the second encounter was used as the first qualifying encounter. Up to ten eligible controls (in terms of sex, age, location) were selected for each case, yielding average 9 controls per case. Each control was also assigned an index date, which is the HFDx date of the matched case. Controls are selected such that they did not meet the HF diagnosis criteria prior to the HFDx date plus 182 days of

| ICD-9 Code | Description |
|---|---|
| 398.91 | Rheumatic heart failure (congestive) |
| 402.01 | Malignant hypertensive heart disease with heart failure |
| 402.11 | Benign hypertensive heart disease with heart failure |
| 402.91 | Unspecified hypertensive heart disease with heart failure |
| 404.01 | Hypertensive heart and chronic kidney disease, malignant, with heart failure and with chronic kidney disease stage I through stage IV, or unspecified |
| 404.03 | Hypertensive heart and chronic kidney disease, malignant, with heart failure and with chronic kidney disease stage V or end stage renal disease |
| 404.11 | Hypertensive heart and chronic kidney disease, benign, with heart failure and with chronic kidney disease stage I through stage IV, or unspecified |
| 404.13 | Hypertensive heart and chronic kidney disease, benign, with heart failure and chronic kidney disease stage V or end stage renal disease |
| 404.91 | Hypertensive heart and chronic kidney disease, unspecified, with heart failure and with chronic kidney disease stage I through stage IV, or unspecified |
| 404.93 | Hypertensive heart and chronic kidney disease, unspecified, with heart failure and chronic kidney disease stage V or end stage renal disease |
| 428.0 | Congestive heart failure, unspecified |
| 428.1 | Left heart failure |
| 428.20 | Systolic heart failure, unspecified |
| 428.21 | Acute systolic heart failure |
| 428.22 | Chronic systolic heart failure |
| 428.23 | Acute on chronic systolic heart failure |
| 428.30 | Diastolic heart failure, unspecified |
| 428.31 | Acute diastolic heart failure |
| 428.32 | Chronic diastolic heart failure |
| 428.33 | Acute on chronic diastolic heart failure |
| 428.40 | Combined systolic and diastolic heart failure, unspecified |
| 428.41 | Acute combined systolic and diastolic heart failure |
| 428.42 | Chronic combined systolic and diastolic heart failure |
| 428.43 | Acute on chronic combined systolic and diastolic heart failure |
| 428.9 | Heart failure, unspecified |

Table 4: Qualifying ICD-9 codes for heart failure

their corresponding case. Control subjects were required to have their first office encounter within one year of the matching HF case patient's first office visit, and have at least one office encounter 30 days before or any time after the case's HFDx date to ensure similar duration of observations among cases and controls.

## C  HYPER-PARAMETER TUNING

We define five hyper-parameters for GRAM:

- dimensionality $m$ of the basic embedding $\mathbf{e}_i$: [100, 200, 300, 400, 500]
- dimensionality $r$ of the RNN hidden layer $\mathbf{h}_t$ from Eq. (4): [100, 200, 300, 400, 500]
- dimensionality $l$ of $\mathbf{W}_a$ and $\mathbf{b}_a$ from Eq. (3): [100, 200, 300, 400, 500]
- $L_2$ regularization coefficient for all weights except RNN weights: [0.1, 0.01, 0.001, 0.0001]
- dropout rate for the dropout on the RNN hidden layer: [0.0, 0.2, 0.4, 0.6, 0.8]

We performed 100 iterations of the random search by using the above ranges for each of the three prediction experiments. For sequential diagnoses prediction on Sutter data, we used 10% of the training data to tune the hyper-parameters to balance the time and search space. To match the baselines' number of parameters to GRAM's, we add 550 to the list of $m$'s possible values. This will make the baseline's largest possible number of parameters comparable to the GRAM's largest possible number of parameters.

For SimpleRollUp and RollUpRare, the number of input codes is smaller than other models due to the grouping. Therefore, to match their largest possible number of parameters to GRAM's, we need to add much larger values to $m$. However, after preliminary experiments, as expected, setting $m$ to too

Table 5: Hyper-parameters used by the models in each predictive modeling experiments

| Experiment | Model | $m$ | $r$ | $l$ | $L_2$ | Dropout rate |
|---|---|---|---|---|---|---|
| Disease progression modeling (Sutter data) | GRAM+ | 500 | 500 | 100 | 0.0001 | 0.6 |
| | GRAM | 500 | 500 | 100 | 0.0001 | 0.6 |
| | RandomDAG | 500 | 500 | 100 | 0.0001 | 0.6 |
| | RNN+ | 550 | 500 | | 0.0001 | 0.6 |
| | RNN | 550 | 500 | | 0.0001 | 0.6 |
| | SimpleRollUp | 500 | 500 | | 0.0001 | 0.4 |
| | RollUpRare | 500 | 500 | | 0.0001 | 0.2 |
| Disease progression modeling (MIMIC-III) | GRAM+ | 400 | 400 | 100 | 0.0001 | 0.6 |
| | GRAM | 400 | 400 | 100 | 0.001 | 0.6 |
| | RandomDAG | 400 | 400 | 100 | 0.001 | 0.6 |
| | RNN+ | 550 | 400 | | 0.001 | 0.8 |
| | RNN | 550 | 400 | | 0.001 | 0.8 |
| | SimpleRollUp | 400 | 400 | | 0.001 | 0.6 |
| | RollUpRare | 400 | 400 | | 0.0001 | 0.0 |
| HF prediction (Sutter HF cohort) | GRAM+ | 200 | 100 | 100 | 0.001 | 0.6 |
| | GRAM | 200 | 100 | 100 | 0.001 | 0.6 |
| | RandomDAG | 300 | 100 | 200 | 0.001 | 0.6 |
| | RNN+ | 200 | 100 | | 0.0001 | 0.6 |
| | RNN | 200 | 100 | | 0.001 | 0.6 |
| | SimpleRollUp | 300 | 200 | | 0.001 | 0.4 |
| | RollUpRare | 100 | 100 | | 0.001 | 0.6 |

large a value degraded the performance due to overfitting. Since the number of input codes decreased due to the grouping, increasing the dimensionality of $\mathbf{e}_i$ is not a logical thing to do. Therefore, for SimpleRollUp and RollUpRare, we use the same list of values for $m$ as other baselines.

Table 5 describes the final hyper-parameter settings we used for all models for each prediction experiments.

## D    PREDICTION RESULTS USING DIFFERENT $k$'S IN ACCURACY@K

We show $Accuracy@k$ using $k = 5, 10, 20, 30$ for sequential diagnoses prediction on Sutter data (Tables 6a, 6b, 6c and 6d) and MIMIC-III (Tables 7a, 7b, 7c and 7d). We can see from the tables that GRAM+ consistently outperforms other models under 40th percentile range, except when $k = 20, 30$ for sequential diagnoses prediction on Sutter data where SimpleRollUp shows similar performance. We can also see that GRAM+ performs significantly better than other models for all $k = 5, 10, 20, 30$ when predicting infrequently observed diseases on MIMIC-III. As discussed in Section 3.2, this seems to come from the short visit sequences of MIMIC patients.

## E    T-SNE 2-D PLOTS OF VARIOUS MODELS

For further comparison, we display t-SNE scatterplots of GRAM (Figure 5a) RandomDAG (Figure 5b, RNN (Figure 5c), and Skip-gram (Figure 5d). GRAM, RandomDAG and RNN were trained for sequential diagnoses prediction on Sutter data, and Skip-gram (Mikolov et al., 2013) was trained on Sutter data as it is an unsupervised method. For Skip-gram, we used each visit $V_t$ as the context window. As we do not distinguish between the target concept and the neighbor concepts, we calculated the Skip-gram objective function using all possible pairs of codes within a single visit.

We can see from Figure 5a that the quality of the final representations $\mathbf{g}_i$ of GRAM is quite similar to GRAM+ (Figure 2a). Compared to other baselines, GRAM demonstrates significantly more structured representations that align well with the given knowledge DAG. It is interesting that Skip-gram shows the most structured representation among all baselines. We used GloVe to initialize the basic embeddings $\mathbf{e}_i$ in this work because it uses global co-occurrence information and its training time is dependent only on the total number of unique concepts $|\mathcal{C}|$. Skip-gram's training time, on the other hand, depends on both the number of patients and the number of visits each patient made, which makes the algorithm generally slower than GloVe. However, considering both Figures 2c and 5d, initializing $\mathbf{e}_i$'s with Skip-gram vectors might give us additional performance boost.

Table 6: Accuracy at various $k$'s (a to d) for sequential diagnoses prediction on Sutter data. The columns represent the labels grouped by the percentile of their frequencies in the training data in non-decreasing order.

| Model | 0-20 | 20-40 | 40-60 | 60-80 | 80-100 |
|---|---|---|---|---|---|
| GRAM+ | **0.0150** | **0.3242** | 0.4325 | 0.4238 | 0.4903 |
| GRAM | 0.0042 | 0.2987 | 0.4224 | 0.4193 | 0.4895 |
| RandomDAG | 0.0050 | 0.2700 | 0.4010 | 0.4059 | 0.4853 |
| RNN+ | 0.0069 | 0.2742 | 0.4140 | 0.4212 | **0.4959** |
| RNN | 0.0080 | 0.2691 | 0.4134 | 0.4227 | 0.4951 |
| SimpleRollUp | 0.0085 | 0.3078 | **0.4369** | **0.4330** | 0.4924 |
| RollUpRare | 0.0062 | 0.2768 | 0.4176 | 0.4226 | 0.4956 |

(a) *Accuracy@5* of sequential diagnoses prediction on Sutter data

| Model | 0-20 | 20-40 | 40-60 | 60-80 | 80-100 |
|---|---|---|---|---|---|
| GRAM+ | **0.0319** | **0.3882** | 0.5054 | 0.5215 | 0.6459 |
| GRAM | 0.0163 | 0.3645 | 0.4944 | 0.5173 | 0.6445 |
| RandomDAG | 0.0142 | 0.3285 | 0.4691 | 0.5025 | 0.6401 |
| RNN+ | 0.0183 | 0.3412 | 0.4884 | 0.5233 | **0.6538** |
| RNN | 0.0196 | 0.3290 | 0.4871 | 0.5230 | 0.6531 |
| SimpleRollUp | 0.0164 | 0.3768 | **0.5132** | **0.5326** | 0.6521 |
| RollUpRare | 0.0204 | 0.3450 | 0.4917 | 0.5228 | 0.6535 |

(b) *Accuracy@10* of sequential diagnoses prediction on Sutter data

| Model | 0-20 | 20-40 | 40-60 | 60-80 | 80-100 |
|---|---|---|---|---|---|
| GRAM+ | **0.0630** | 0.4486 | 0.5764 | 0.6153 | 0.7973 |
| GRAM | 0.0442 | 0.4276 | 0.5669 | 0.6125 | 0.7963 |
| RandomDAG | 0.0397 | 0.3933 | 0.5389 | 0.5997 | 0.7919 |
| RNN+ | 0.0483 | 0.4132 | 0.5654 | 0.6235 | 0.8003 |
| RNN | 0.0481 | 0.4025 | 0.5630 | 0.6232 | 0.7995 |
| SimpleRollUp | 0.0418 | **0.4496** | **0.5877** | **0.6262** | **0.8013** |
| RollUpRare | 0.0517 | 0.4170 | 0.5672 | 0.6214 | 0.8002 |

(c) *Accuracy@20* of sequential diagnoses prediction on Sutter data

| Model | 0-20 | 20-40 | 40-60 | 60-80 | 80-100 |
|---|---|---|---|---|---|
| GRAM+ | **0.0946** | 0.4879 | 0.6186 | 0.6792 | **0.8800** |
| GRAM | 0.0662 | 0.4693 | 0.6107 | 0.6766 | 0.8798 |
| RandomDAG | 0.0672 | 0.4313 | 0.5843 | 0.6667 | 0.8760 |
| RNN+ | 0.0736 | 0.4604 | 0.6136 | **0.6930** | 0.8785 |
| RNN | 0.0733 | 0.4478 | 0.6103 | 0.6921 | 0.8767 |
| SimpleRollUp | 0.0662 | **0.4924** | **0.6312** | 0.6907 | 0.8795 |
| RollUpRare | 0.0759 | 0.4657 | 0.6146 | 0.6908 | 0.8778 |

(d) *Accuracy@30* of sequential diagnoses prediction on Sutter data

Table 7: Accuracy at various $k$'s (a to d) for sequential diagnoses prediction on MIMIC-III. The columns represent the labels grouped by the percentile of their frequencies in the training data in non-decreasing order.

| Model | 0-20 | 20-40 | 40-60 | 60-80 | 80-100 |
|---|---|---|---|---|---|
| GRAM+ | **0.0086** | **0.1089** | **0.1665** | 0.1029 | 0.2597 |
| GRAM | 0.0000 | 0.0468 | 0.1093 | 0.0918 | **0.2665** |
| RandomDAG | 0.0000 | 0.0327 | 0.0778 | 0.0612 | 0.1634 |
| RNN+ | 0.0000 | 0.0435 | 0.1266 | 0.0973 | 0.2594 |
| RNN | 0.0000 | 0.0376 | 0.1105 | 0.0923 | 0.2601 |
| SimpleRollUp | 0.0000 | 0.0671 | 0.1501 | **0.1191** | 0.2635 |
| RollUpRare | 0.0000 | 0.0423 | 0.1085 | 0.0874 | 0.2604 |

(a) *Accuracy@5* of sequential diagnoses prediction on MIMIC-III

| Model | 0-20 | 20-40 | 40-60 | 60-80 | 80-100 |
|---|---|---|---|---|---|
| GRAM+ | **0.0380** | **0.1310** | **0.2095** | 0.1627 | 0.4175 |
| GRAM | 0.0045 | 0.0682 | 0.1494 | 0.1487 | 0.4235 |
| RandomDAG | 0.0023 | 0.0470 | 0.1025 | 0.0938 | 0.2692 |
| RNN+ | 0.0227 | 0.0587 | 0.1591 | 0.1616 | 0.4193 |
| RNN | 0.0023 | 0.0518 | 0.1389 | 0.1521 | 0.4142 |
| SimpleRollUp | 0.0249 | 0.1038 | 0.1997 | **0.1769** | **0.4260** |
| RollUpRare | 0.0227 | 0.0530 | 0.1412 | 0.1519 | 0.4180 |

(b) *Accuracy@10* of sequential diagnoses prediction on MIMIC-III

| Model | 0-20 | 20-40 | 40-60 | 60-80 | 80-100 |
|---|---|---|---|---|---|
| GRAM+ | **0.0672** | **0.1787** | **0.2644** | 0.2490 | 0.6267 |
| GRAM | 0.0556 | 0.1016 | 0.1935 | 0.2296 | 0.6363 |
| RandomDAG | 0.0329 | 0.0708 | 0.1346 | 0.1512 | 0.4494 |
| RNN+ | 0.0454 | 0.0843 | 0.2080 | 0.2494 | 0.6239 |
| RNN | 0.0454 | 0.0731 | 0.1804 | 0.2371 | 0.6243 |
| SimpleRollUp | 0.0578 | 0.1328 | 0.2455 | **0.2667** | **0.6387** |
| RollUpRare | 0.0454 | 0.0653 | 0.1843 | 0.2364 | 0.6277 |

(c) *Accuracy@20* of sequential diagnoses prediction on MIMIC-III

| Model | 0-20 | 20-40 | 40-60 | 60-80 | 80-100 |
|---|---|---|---|---|---|
| GRAM+ | **0.0744** | **0.2065** | **0.3180** | 0.3363 | 0.7726 |
| GRAM | 0.0578 | 0.1157 | 0.2257 | 0.3074 | **0.7802** |
| RandomDAG | 0.0351 | 0.0932 | 0.1635 | 0.2200 | 0.5977 |
| RNN+ | 0.0578 | 0.1103 | 0.2571 | 0.3409 | 0.7656 |
| RNN | 0.0578 | 0.0775 | 0.2237 | 0.3160 | 0.7643 |
| SimpleRollUp | 0.0578 | 0.1556 | 0.2865 | **0.3488** | 0.7800 |
| RollUpRare | 0.0556 | 0.0910 | 0.2235 | 0.3255 | 0.7661 |

(d) *Accuracy@30* of sequential diagnoses prediction on MIMIC-III

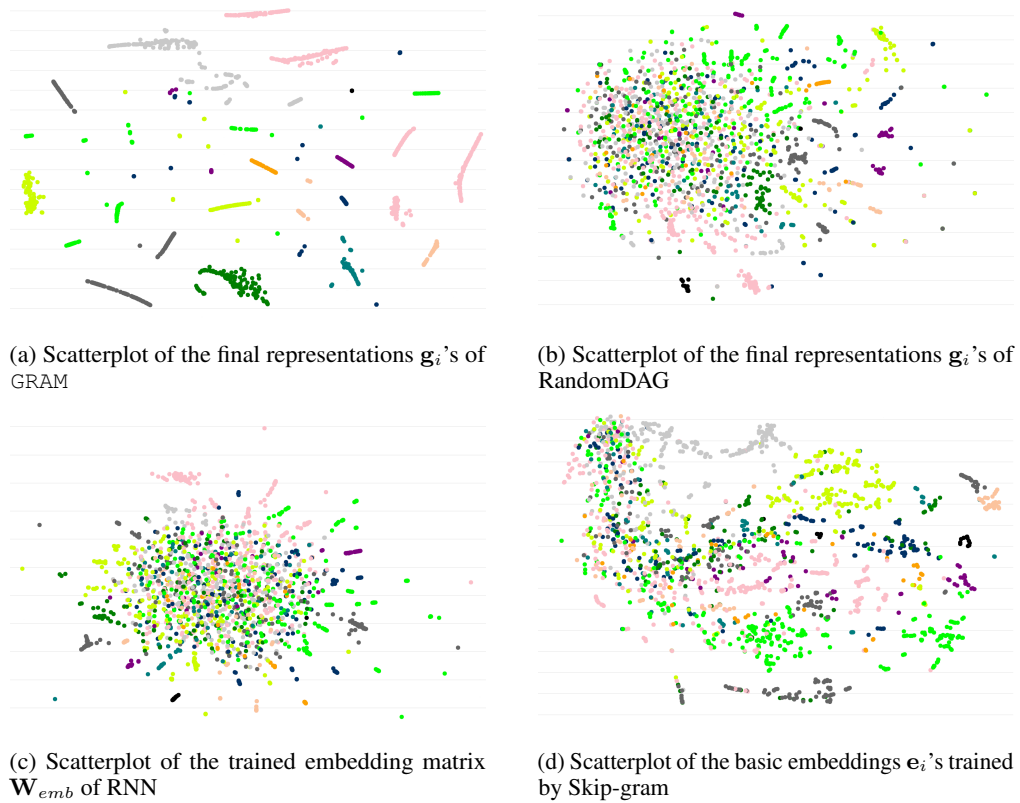

(a) Scatterplot of the final representations $\mathbf{g}_i$'s of `GRAM`

(b) Scatterplot of the final representations $\mathbf{g}_i$'s of RandomDAG

(c) Scatterplot of the trained embedding matrix $\mathbf{W}_{emb}$ of RNN

(d) Scatterplot of the basic embeddings $\mathbf{e}_i$'s trained by Skip-gram

Figure 5: Scatterplot of medical concepts trained by various models. We used t-SNE to reduce the dimension to 2-D.

