# Peer review of "GRAM: Graph-based Attention Model for Healthcare Representation Learning"

_ICLR 2017 — rejected_

[Official Review · AnonReviewer3 · rating 6 · confidence 4 · 16 Dec 2016 (modified: 20 Jan 2017)]
**Interesting approach for learning input representations in RNN**

I read the authors' response and maintain my rating.

---

This paper introduces an approach for integrating a direct acyclic graph structure of the data into word / code embeddings, in order to leverage domain knowledge and thus help train an RNN with scarce data. It is applied to codes of medical visits. Each code is part of an ontology, which can be represented by a DAG, where codes correspond to leaf nodes, and where different codes may share common ancestors (non-leaf nodes) in the DAG. Instead of embedding merely the leaf nodes, one can also embed the non-leaf nodes, and the embeddings of the code and its ancestors can be combined using a convex sum. That convex sum can be seen as an attention mechanism over the representation. The attention weights depend on the embeddings and the weights of an MLP, meaning that the model can separate learning the code embeddings and the interaction between the codes. Embedding codes are pretrained using GloVe, then fine-tuned.

The model is properly evaluated on two medical datasets, with several variations to isolate the contribution of the DAG (GRAM or GRAM+ vs. RNN or RandomDAG) and of pretraining the embeddings (RNN+ vs RNN, GRAM+ vs GRAM). Both are shown to help achieve the best performance and the evaluation methodology seems thorough.

The paper is also well written, and the case for MLP attention instead of a plain dot product of embeddings was made by the authors.

My only two comments would be:
1) Why is there a softmax in equation 4, given that the loss is multivariate cross-entropy (in the predicted visit, several codes could be equal to 1), not a a single-class cross-entropy?
2) What is the embedding dimension m?

[Official Review · AnonReviewer2 · rating 6 · confidence 3 · 16 Dec 2016]
**No Title**

SUMMARY.
This paper presents a method for enriching medical concepts with their parent nodes in an ontology.
The method employs an attention mechanism over the parent nodes of a medical concept to create a richer representation of the concept itself.
The rationale of this is that for  infrequent medical concepts the attention mechanism will rely more on general concepts, higher in the ontology hierarchy, while for frequent ones will focus on the specific concept.
The attention mechanism is trained together with a recurrent neural network and the model accuracy is tested on two tasks.
The first task aims at prediction the diagnosis categories at each time step, while the second task aims at predicting whether or not a heart failure is likely to happen after the T-th step.

Results shows that the proposed model works well in condition of data insufficiency.

----------

OVERALL JUDGMENT
The proposed model is simple but interesting.
The ideas presented are worth to expand but there are also some points where the authors could have done better.
The learning of the representation of concepts in the ontology is a bit naive, for example the authors could have used some kind of knowledge base factorization approach to learn the concepts, or some graph convolutional approach.
I do not see why the the very general factorization methods for knowledge bases do not apply in the case of ontology learning.
I also found strange that the representation of leaves are fine tuned while the inner nodes are not, it is a specific reason to do so?

Regarding the presentation, the paper is clear and the qualitative evaluation is insightful.


----------

DETAILED COMMENTS

Figure 2. Please use the same image format with the same resolution.

[Public Comment · (anonymous) · 22 Dec 2016]
**How that compares to your KDD 2016 paper**

Nice paper, I have one comment however. 

I read your KDD 2016 paper about "Multi-layer Representation Learning for Medical Concepts" where you presented a Med2vec model to map patients' visits and codes into a different space. You showed that the new representation is effective and interpretable. 

Your current submission to ICLR is very similar to your KDD paper except that you have regularized the model using DAG of concepts as a prior knowledge. 
1) Am I right on that? 
2) why you did not compare your current model to Med2vec? You can use Med2vec as input to any other model such as RNN or CNN for classification. This case, we can have a clear picture about the contribution of the prior knowledge to the model accuracy. If there is no significant difference in the results then why we would care about the prior knowledge.

[Official Review · AnonReviewer1 · rating 6 · confidence 4 · 23 Dec 2016]
**No Title**

This paper addresses the problem of data sparsity in the healthcare domain by leveraging hierarchies of medical concepts organized in ontologies. The paper focuses on sequential prediction given a patient’s medical record (a sequence of medical codes, some of which might occur very rarely). Instead of simply assigning each medical code an independent embedding before feeding it to an RNN, the proposed approach assigns each node in the medical ontology a “basic” embedding, and composes a “final” embedding for each medical code by taking a learned weighted average (via an attention mechanism) of the medical code’s ancestors in the ontology. Notably, the paper is well written and the approach is quite intuitive.
I have the following comments: 
- Why is the patient’s visit taken as just the sum of medical codes found in the visit, and not say the average or a learned weighted average? Wouldn’t this bias for/against the number of codes in the visit?
- I don’t see why basic embeddings are not fine tuned as well. Did you find that to hurt performance? Do you have an explanation for that?
- Looking at Figure 2, the results seem very close and the figures are not very clear (figure (b) top is missing). Also, I am wondering how significant the differences are so it would be nice to comment on that.
Finally, I think this is an interesting application paper applying well-established deep learning techniques. The paper deals with an important issue that arises when applying deep learning models in domains with scarce data resources. However, I would like the authors to comment on what there paper offers as new insights to the ICLR community and why they think ICLR is a good avenue for their work.

[Author Response · Edward Choi · 12 Jan 2017]
**Changes in the last revision**

Major changes in the revision
-- Figures 2 are replaced with tables for better readability
-- Related works have been expanded
-- Incorporated the reviewers’ comments in the experiment section.
-- Experiments for GRAM+ and RNN+ in heart failure prediction were re-run

More details
We discovered that, for GRAM+ and RNN+ in HF prediction with varying amounts of training data, we initialized the basic embeddings e_i’s and the embedding matrix W_emb with GloVe vectors that were trained always on the 100% training data instead of the downsampled training data. This could have exaggerated the AUC of both models. Therefore we conducted a correct experiment where the initialization was done with GloVe vectors trained on the downsampled training data. We found that GRAM+’s performance did not show noticeable difference, but RNN+’s performance dropped approximately 0.5-1% AUC.

[Final Decision · Program Chairs · 06 Feb 2017]
**ICLR committee final decision**

The reviewers all agreed that the paper was well written, that the proposed approach is very sensible and intuitive and that the experiments are convincing. However, they are concerned that the proposed work is of limited interest to the ICLR community. The technical contribution is not significant enough for any of the reviewers to strongly recommend an acceptance.